# Constraining Variational Inference with Geometric Jensen-Shannon Divergence

**Jacob Deasy**[*]**, Nikola Simidjievski, Pietro Liò**
Department of Computer Science and Technology
University of Cambridge
{jd645,ns779,pl219}@cam.ac.uk

## Abstract

We examine the problem of *controlling divergences* for latent space regularisation in variational autoencoders. Specifically, when aiming to reconstruct example $x \in \mathbb{R}^m$ via latent space $z \in \mathbb{R}^n$ $(n \leq m)$, while balancing this against the need for generalisable latent representations. We present a regularisation mechanism based on the *skew-geometric Jensen-Shannon divergence* $\left(\text{JS}^{\text{G}_\alpha}\right)$. We find a variation in $\text{JS}^{\text{G}_\alpha}$, motivated by limiting cases, which leads to an intuitive interpolation between forward and reverse KL in the space of both distributions and divergences. We motivate its potential benefits for VAEs through low-dimensional examples, before presenting quantitative and qualitative results. Our experiments demonstrate that skewing our variant of $\text{JS}^{\text{G}_\alpha}$, in the context of $\text{JS}^{\text{G}_\alpha}$-VAEs, leads to better reconstruction and generation when compared to several baseline VAEs. Our approach is entirely unsupervised and utilises only one hyperparameter which can be easily interpreted in latent space.

## 1 Introduction

The problem of controlling regularisation strength for generative models is often data-dependent and poorly understood [3, 7]. Post-hoc analysis of coefficients dictating regularisation strength is rarely carried out and even more rarely provides an intuitive explanation (e.g. $\beta$-VAE, [13]). Although evidence suggests that stronger regularisation in variational settings leads to desirable disentangled representations of latent factors and better generalisation [38], scaling factors remain opaque and unrelated to the task at hand.

To learn useful latent representations for reconstruction and generation of high-dimensional distributions, the variational inference problem can be addressed through the use of Variational Autoencoders (VAEs) [17, 34]. VAE learning requires optimisation of an objective balancing the quality of samples that are encoded and then decoded, with a regularisation term penalising latent space deviations from a fixed prior distribution. VAEs have favourable properties when compared with other families of generative models, such as Generative Adversarial Networks (GANs) [10] and autoregressive models [9, 20]. In particular, GANs are known to necessitate more stringent and problem-dependent training regimes, while autoregressive models are computationally expensive and inefficient to sample.

VAEs often assume latent variables to be parameterised by a multivariate Gaussian $p_\theta(z) = N(\mu, \sigma^2)$ with $z, \mu, \sigma \in \mathbb{R}^n$, which is approximated by $q_\phi(z|x)$ with $x \in \mathbb{R}^m$ and $n \leq m$. In variational Bayesian methods, using the Evidence Lower BOund (ELBO) [4], the model can be naturally constrained to prevent overfitting by minimising the Kullback-Leibler (KL) [19] divergence to an isotropic unit Gaussian ball $\text{KL}\left(p_\theta(z) \parallel \mathcal{N}(0, I)\right)$. One line of work has sought to better understand this divergence term to induce disentanglement, robustness, and generalisation [5, 6]. Meanwhile, the

---
[*]Corresponding author.

broader framework of learning a VAE as a constrained optimisation problem [13], has allowed for increasing use of more exotic statistical divergences and distances for latent space regularisation [8, 12, 22, 37], such as the regularisation term in InfoVAE [38], the Maximum Mean Discrepancy (MMD) [11].

As regularisation terms increase in complexity, it is advantageous to maintain intuition as to how they operate in latent space and to avoid exponential hyperparameter search spaces on real-world problems. In order to properly capitalise on the advantages of each divergence, it is also desirable that the meaning of scaling factors remains clear when combining multiple divergence terms. For instance, as forward KL and reverse KL are known to have distinct beneficial properties—*zero-avoidance* allowing for exploration of new areas in the latent space [3] and *zero-forcing* more easily ignoring noise for sharper selection of strong modes [37] respectively—there are instances where favouring one over the other would be beneficial. Even better would be to balance the use of both properties at the same time in a comprehensible manner.

In this regard, we propose the *skew-geometric Jensen-Shannon Variational Autoencoder* ($\text{JS}^{\text{G}_\alpha}$-VAE) as an unsupervised approach to learning strongly regularised latent spaces. More specifically, we make several contributions: we first discuss the skew-geometric Jensen-Shannon divergence (and its dual form) [30] in the context of the well known KL and Jensen-Shannon (JS) divergences and outline its limited use. We proceed to propose an adjustment of the skew parameter, and show how its effect on an intermediate distribution in $\text{JS}^{\text{G}_\alpha}$ furnishes us with a more intuitive divergence and permits interpolation between forward and reverse KL divergence. We then study the skew-geometric Jensen-Shannon in the wider context of latent space regularisation and use it to derive a loss function for $\text{JS}^{\text{G}_\alpha}$-VAE.

To test the utility of the proposed skew-geometric Jensen-Shannon adjustments, we investigate how $\text{JS}^{\text{G}_\alpha}$ operates on low-dimensional examples. We demonstrate that $\text{JS}^{\text{G}_\alpha}$ has beneficial properties for light-tailed posterior distributions and is a more useful (and tractable) intermediate divergence than standard JS. We further exhibit that $\text{JS}^{\text{G}_\alpha}$ for VAEs has a positive impact on test set reconstruction loss. Namely, we show that the dual form, $\text{JS}^{\text{G}_\alpha}_*$ consistently outperforms forward and reverse KL across several standard benchmark datasets and skew values.[2]

## 2 $\text{JS}^{\text{G}_\alpha}$ VAE derivation

Existing work suggests that there exists no tractable interpolation between forward and reverse KL for multivariate Gaussians. In this section, we will show that one can be found by adapting $\text{JS}^{\text{G}_\alpha}$. We also exhibit how this interpolation, well-motivated in the space of distributions, reduces to a simple quadratic interpolation in the space of divergences.

### 2.1 The $\text{JS}^{\text{G}_\alpha}$ divergences family

**Problems with KL and JS minimisation.** For distributions $P$ and $Q$ of a continuous random variable $X = [X_1, \ldots, X_n]^{\text{T}}$, the Kullback-Leibler (KL) divergence [19] is defined as

$$\text{KL}(P \parallel Q) = \int_X p(x) \log \left[ \frac{p(x)}{q(x)} \right] dx, \tag{1}$$

where $p$ and $q$ are the probability densities of $P$ and $Q$ respectively, and $x \in \mathbb{R}^n$. In particular, Equation (1) is known as the forward KL divergence from $P$ to $Q$, whereas reverse KL divergence refers to $\text{KL}(Q \parallel P)$.

Due to Gaussian distributions being the self-conjugate distributions of choice in variational learning, we are interested in using divergences to compare two multivariate normal distributions $\mathcal{N}_1(\mu_1, \Sigma_1)$ and $\mathcal{N}_2(\mu_2, \Sigma_2)$ with the same dimension $n$. In this case, the KL divergence is

$$\text{KL}(\mathcal{N}_1 \parallel \mathcal{N}_2) = \frac{1}{2} \left( \text{tr}\left( \Sigma_2^{-1} \Sigma_1 \right) + \ln \left[ \frac{|\Sigma_2|}{|\Sigma_1|} \right] + (\mu_2 - \mu_1)^{\text{T}} \Sigma_2^{-1} (\mu_2 - \mu_1) - n \right). \tag{2}$$

This expression is well-known in variational inference and, for the case of reverse KL from a standard normal distribution $\mathcal{N}_2(0, I)$ to a diagonal multivariate normal distribution, reduces to the expression

$$\mathrm{KL}\left(\mathcal{N}_1\left(\mu_1, \mathrm{diag}\left(\sigma_1^2, \ldots, \sigma_n^2\right)\right) \| \mathcal{N}_2(0, I)\right) = \frac{1}{2}\sum_{i=1}^n\left(\sigma_i^2 - \ln\left[\sigma_i^2\right] + \mu_i^2 - 1\right), \qquad (3)$$

used as a regularisation term in variational models [13, 17, 27] and is known to enforce zero-avoiding parameters on $\mathcal{N}_1$ when minimised [3, 26]. On the other hand, the forward KL divergence reduces to

$$\mathrm{KL}\left(\mathcal{N}_2(0, I) \| \mathcal{N}_1\left(\mu_1, \mathrm{diag}\left(\sigma_1^2, \ldots, \sigma_n^1\right)\right)\right) = \frac{1}{2}\sum_{i=1}^n\left(\sigma_i^{-2} + \ln\left[\sigma_i^2\right] + \frac{\mu_i^2}{\sigma_i^2} - 1\right), \qquad (4)$$

and is known for its zero-forcing property [3, 26]. However, there exist well-known drawbacks of the KL divergence, such as no upper bound leading to unstable optimization and poor approximation [12], as well as its asymmetric property $\mathrm{KL}(P \| Q) \neq \mathrm{KL}(Q \| P)$. Underdispersed approximations relative to the exact posterior also produce difficulties with light-tailed posteriors when the variational distribution has heavier tails [8].

One attempt at remedying these issues is the well-known symmetrisation, the Jensen-Shannon (JS) divergence [23]

$$\mathrm{JS}(p(z) \| q(x)) = \frac{1}{2}\mathrm{KL}\left(p \,\Big\|\, \frac{p+q}{2}\right) + \frac{1}{2}\mathrm{KL}\left(q \,\Big\|\, \frac{p+q}{2}\right). \qquad (5)$$

Although the JS divergence is bounded (in $[0, 1]$ when using base 2), and offers some intuition through symmetry, it includes the problematic mixture distribution $\frac{p+q}{2}$. This term means that no closed-form expression exists for the JS divergence between two multivariate normal distributions using Equation (5).

**Divergence families.** To circumvent these problems, prior work has sought more general families of distribution divergence [29]. For example, when $\lambda = \frac{1}{2}$, JS is a special case of the more general family of $\lambda$ divergences, defined by

$$\lambda(p(x) \| q(x)) = \lambda\mathrm{KL}\left(p \| (1-\lambda)p + \lambda q\right) + (1-\lambda)\mathrm{KL}\left(q \| (1-\lambda)p + \lambda q\right), \qquad (6)$$

for $\lambda \in [0, 1]$, which interpolates between forward and reverse KL, and provides control over the degree of *divergence skew* (how closely related the intermediate distribution is to $p$ or $q$).

Although $\lambda$ divergences do not prevent the intractable comparison to a mixture distribution, their broader goal is to measure weighted divergence to an intermediate distribution in the space of possible distributions over $X$. In the case of the JS divergence, this is the (arithmetic) mean divergence to the arithmetic mean distribution. Recently, [30] and [32] have proposed a further generalisation of the JS divergence using abstract means (*quasi-arithmetic means* [28], also known as *Kolmogorov-Nagumo means*). By choosing the *weighted geometric mean* $G_\alpha(x, y) = x^{1-\alpha}y^\alpha$ for $\alpha \in [0, 1]$, and using the property that the weighted product of exponential family distributions (which includes the multivariate normal) stays in the exponential family [31], a new divergence family has arisen

$$\mathrm{JS}^{G_\alpha}(p(x) \| q(x)) = (1-\alpha)\mathrm{KL}\left(p \| G_\alpha(p, q)\right) + \alpha\mathrm{KL}\left(q \| G_\alpha(p, q)\right). \qquad (7)$$

$\mathrm{JS}^{G_\alpha}$, the *skew-geometric Jensen-Shannon divergence*, between two multivariate Gaussians $\mathcal{N}(\mu_1, \Sigma_1)$ and $\mathcal{N}(\mu_2, \Sigma_2)$ then admits the closed form

$$\mathrm{JS}^{G_\alpha}\left(\mathcal{N}_1 \| \mathcal{N}_2\right) = (1-\alpha)\mathrm{KL}\left(\mathcal{N}_1 \| \mathcal{N}_\alpha\right) + \alpha\mathrm{KL}\left(\mathcal{N}_2 \| \mathcal{N}_\alpha\right) \qquad (8)$$

$$= \frac{1}{2}\left(\mathrm{tr}\left(\Sigma_\alpha^{-1}((1-\alpha)\Sigma_1 + \alpha\Sigma_2)\right) + \log\left[\frac{|\Sigma_\alpha|}{|\Sigma_1|^{1-\alpha}|\Sigma_2|^\alpha}\right]\right.$$

$$\left. + (1-\alpha)(\mu_\alpha - \mu_1)^\mathrm{T}\Sigma_\alpha^{-1}(\mu_\alpha - \mu_1) + \alpha(\mu_\alpha - \mu_2)^\mathrm{T}\Sigma_\alpha^{-1}(\mu_\alpha - \mu_2) - n\right), \quad (9)$$

with the equivalent dual divergence being

$$\mathrm{JS}_*^{G_\alpha}\left(\mathcal{N}_1 \| \mathcal{N}_2\right) = (1-\alpha)\mathrm{KL}\left(\mathcal{N}_\alpha \| \mathcal{N}_1\right) + \alpha\mathrm{KL}\left(\mathcal{N}_\alpha \| \mathcal{N}_2\right) \qquad (10)$$

$$= \frac{1}{2}\left((1-\alpha)\mu_1^\mathrm{T}\Sigma_1^{-1}\mu_1 + \alpha\mu_2^\mathrm{T}\Sigma_2^{-1}\mu_2 - \mu_\alpha^\mathrm{T}\Sigma_\alpha^{-1}\mu_\alpha + \log\left[\frac{|\Sigma_1|^{1-\alpha}|\Sigma_2|^\alpha}{|\Sigma_\alpha|}\right]\right),$$

$$(11)$$

where $\mathcal{N}_\alpha$ has parameters

$$\Sigma_\alpha = \left((1-\alpha)\Sigma_1^{-1} + \alpha\Sigma_2^{-1}\right)^{-1}, \tag{12}$$

(the matrix harmonic barycenter) and

$$\mu_\alpha = \Sigma_\alpha \left((1-\alpha)\Sigma_1^{-1}\mu_1 + \alpha\Sigma_2^{-1}\mu_2\right). \tag{13}$$

Throughout this paper we explore how to incorporate these expressions into variational learning.

## 2.2 $\mathrm{JS}^{\mathrm{G}_\alpha}$ and $\mathrm{JS}^{\mathrm{G}_\alpha}_*$ in variational neural networks

**Interpolation between forward and reverse KL.**  Before applying $\mathrm{JS}^{\mathrm{G}_\alpha}$, we note that although the mean distribution $\mathcal{N}_\alpha$ can be intuitively understood, the limiting skew cases still seem to offer no insight, as

$$\lim_{\alpha \to 0}\left[\mathrm{JS}^{\mathrm{G}_\alpha}\right] = 0 \qquad\qquad \lim_{\alpha \to 1}\left[\mathrm{JS}^{\mathrm{G}_\alpha}\right] = 0 \tag{14}$$

$$\lim_{\alpha \to 0}\left[\mathrm{JS}^{\mathrm{G}_\alpha}_*\right] = 0 \qquad\qquad \lim_{\alpha \to 1}\left[\mathrm{JS}^{\mathrm{G}_\alpha}_*\right] = 0. \tag{15}$$

Therefore, we instead choose to consider the more useful intermediate mean distribution

$$\mathcal{N}_{\alpha'} = \mathcal{N}\left(\mu_{(1-\alpha)}, \Sigma_{(1-\alpha)}\right). \tag{16}$$

This, is equivalent to simply reversing the geometric mean (using $G_\alpha(y, x)$ rather than $G_\alpha(x, y)$) and trivially still permits a valid divergence as a weighted sum of valid divergences.

**Proposition 1.** *The alternative divergence*

$$\mathrm{JS}^{\mathrm{G}_{\alpha'}}\left(\mathcal{N}_1 \parallel \mathcal{N}_2\right) = (1-\alpha)\mathrm{KL}\left(\mathcal{N}_1 \parallel \mathcal{N}_{\alpha'}\right) + \alpha\mathrm{KL}\left(\mathcal{N}_2 \parallel \mathcal{N}_{\alpha'}\right), \tag{17}$$

*and its dual* $\mathrm{JS}^{\mathrm{G}_{\alpha'}}_*$*, interpolate between forward and reverse* KL*, satisfying*

$$\lim_{\alpha \to 0}\left[\mathrm{JS}^{\mathrm{G}_{\alpha'}}\right] = \mathrm{KL}\left(\mathcal{N}_1 \parallel \mathcal{N}_2\right) \qquad\qquad \lim_{\alpha \to 1}\left[\mathrm{JS}^{\mathrm{G}_{\alpha'}}\right] = \mathrm{KL}\left(\mathcal{N}_2 \parallel \mathcal{N}_1\right) \tag{18}$$

$$\lim_{\alpha \to 0}\left[\mathrm{JS}^{\mathrm{G}_{\alpha'}}_*\right] = \mathrm{KL}\left(\mathcal{N}_2 \parallel \mathcal{N}_1\right) \qquad\qquad \lim_{\alpha \to 1}\left[\mathrm{JS}^{\mathrm{G}_{\alpha'}}_*\right] = \mathrm{KL}\left(\mathcal{N}_1 \parallel \mathcal{N}_2\right). \tag{19}$$

The proof of this is given in Appendix A.1. Note that this is a special case of Definition 5 in [30]. Henceforth in the paper, unless *explicitly* stated, $\mathrm{JS}^{\mathrm{G}_\alpha}$ refers to $\mathrm{JS}^{\mathrm{G}_{\alpha'}}$ (without the prime ($'$)).

**Variational autoencoders.**  We can now introduce a new VAE loss function based on this finding by using the formulation of VAE optimisation as a constrained optimisation problem given in [13]. For generative models, a suitable objective to maximise is the marginal (log-)likelihood of the observed data $x \in \mathbb{R}^m$ as an expectation over the whole distribution of latent factors $z \in \mathbb{R}^n$

$$\max_\theta \left[\mathbb{E}_{p_\theta(z)}\left[p_\theta(x|z)\right]\right]. \tag{20}$$

More generalisable latent representations can be achieved by imposing an isotropic unit Gaussian constraint on the prior $p(z) = \mathcal{N}(0, I)$, arriving at the constrained optimisation problem

$$\max_{\phi,\theta} \mathbb{E}_{p_\mathcal{D}(x)}\left[\log \mathbb{E}_{q_\phi(z|x)}\left[p_\theta(x|z)\right]\right] \qquad\qquad \text{subject to } D(q_\phi(z|x) \parallel p(z)) < \varepsilon, \tag{21}$$

where $\varepsilon$ dictates the strength of the constraint and $D$ is a divergence. We can then re-write Equation (21) as a Lagrangian under the KKT conditions [15, 18], obtaining

$$\mathcal{F}(\theta, \phi, \lambda; x, z) = \mathbb{E}_{q_\phi(z|x)}\left[\log p_\theta(x|z)\right] - \lambda\left(D(q_\phi(z|x) \parallel p(z)) - \varepsilon\right). \tag{22}$$

By setting $D(\alpha) = \mathrm{JS}^{\mathrm{G}_\alpha}$ or $D(\alpha) = \mathrm{JS}^{\mathrm{G}_\alpha}_*$, we immediately note that our family of divergences includes the $\beta$-VAE by setting $\alpha = 1$ and varying $\lambda$. In simple terms, a broader family of divergences using both $\alpha$ and $\beta$, would dictate *where* and with how much strength to skew an intermediate distribution.

Before experimentation, in order to use $\mathrm{JS}^{\mathrm{G}_\alpha}$ and $\mathrm{JS}^{\mathrm{G}_\alpha}_*$ as divergence measures in variational learning, we first simplify Equations (9) and (11).

**Proposition 2.** *For a diagonal multivariate normal distribution $\mathcal{N}_1(\mu, diag\left(\sigma_1^2, \ldots, \sigma_n^1\right))$ and a standard normal distribution $\mathcal{N}_2(0, I)$, the skew-geometric Jensen-Shannon divergence $\mathrm{JS}^{\mathrm{G}_\alpha}$—an intermediate of forward and reverse* KL *regularisation—and its dual $\mathrm{JS}^{\mathrm{G}_\alpha}_*$ reduce to*

$$\mathrm{JS}^{\mathrm{G}_\alpha}(\mathcal{N}_1 \parallel \mathcal{N}_2) = \frac{1}{2}\sum_{i=1}^{n}\left(\frac{(1-\alpha)\sigma_i^2 + \alpha}{\sigma_{\alpha,i}^2} + \log\left[\frac{\sigma_{\alpha,i}^2}{\sigma_i^{2(1-\alpha)}}\right] + \frac{(1-\alpha)(\mu_{\alpha,i} - \mu_i)^2}{\sigma_{\alpha,i}^2} + \frac{\alpha\mu_{\alpha,i}^2}{\sigma_{\alpha,i}^2} - 1\right),$$
(23)

*and*

$$\mathrm{JS}^{\mathrm{G}_\alpha}_* = \frac{1}{2}\sum_{i=1}^{n}\left(\frac{\mu_i^2}{\sigma_i^2} - \frac{\mu_{\alpha,i}^2}{\sigma_\alpha^2} + \log\left[\frac{\sigma_i^{2(1-\alpha)}}{\sigma_{\alpha,i}^2}\right]\right),$$
(24)

*respectively, where*

$$\sigma_{\alpha,i}^2 = \frac{\sigma_i^2}{(1-\alpha) + \alpha\sigma_i^2},$$
(25)

*and*

$$\mu_{\alpha,i} = \frac{\sigma_{\alpha,i}^2(1-\alpha)\mu_i}{\sigma_i^2}.$$
(26)

The proof of this is given in Appendix A.2.

## 3 Experiments

Thus far we have discussed the $\mathrm{JS}^{\mathrm{G}_\alpha}$ divergence and its relationship to KL and in particular VAEs. In this section, we begin by offering a better understanding of where $\mathrm{JS}^{\mathrm{G}_\alpha}$ and its variants differ in distributional space. We then provide a quantitative and qualitative exploration, justifying the immediate benefit of skewing $\alpha$ away from 0 or 1, before finishing with an exploration of the effects this has on VAE reconstruction as well as on the generative capabilities. Note that, in the analyses that follow, we set $\lambda = 1$ for all variants of $\mathrm{JS}^{\mathrm{G}_\alpha}$-VAEs[3].

### 3.1 Characteristic behaviour of $\mathrm{JS}^{\mathrm{G}_\alpha}$

To elucidate how $\mathrm{JS}^{\mathrm{G}_\alpha}$ will behave in the higher dimensional setting of variational inference, we highlight its properties in the case of one and two dimensions. In Figure 1, univariate Gaussians illustrate how the integrand for $\mathrm{JS}^{\mathrm{G}_\alpha}$ differs favourably from the intractable JS. As the intermediate distribution $\mathcal{N}_\alpha$ in Figure 1a is a Gaussian, $\mathrm{JS}^{\mathrm{G}_\alpha}$ not only permits a closed-form integral, but also offers a more natural interpolation between $p(z)$ and $q(z|x)$, which raises questions about whether intuitive regularisation strength (relative to a known intermediate Gaussian) may be possible in variational settings. Moreover, Figure 1c demonstrates symmetry for $\alpha = 0.5$, and both Figure 1b and Figure 1c depict the increased integrand in areas of low probability density—addressing the issues touched upon earlier, where KL struggles with light-tailed posteriors.

In Figure 2, we use two dimensions to depict the effect of changing divergence measures on optimisation. As the integral of JS divergence is not tractable (and to make comparison fair), we directly optimise a bivariate Gaussian via samples from the data for all divergences. We see that the example mixture of Gaussians leads to the zero-avoiding property of KL divergence in Figure 2a and zero-forcing (i.e. mode dropping) for reverse KL in Figure 2b. While JS divergence provides an intermediate solution in Figure 2c, there is still considerable unnecessary spreading and direct optimisation of the integral will not scale. Finally, $\mathrm{JS}^{\mathrm{G}_\alpha}$ with $\alpha$ naively set to the symmetric case $\alpha = 0.5$ leads to a more reasonable intermediate distribution which both tends towards the dominant mode and offers localised exploration.

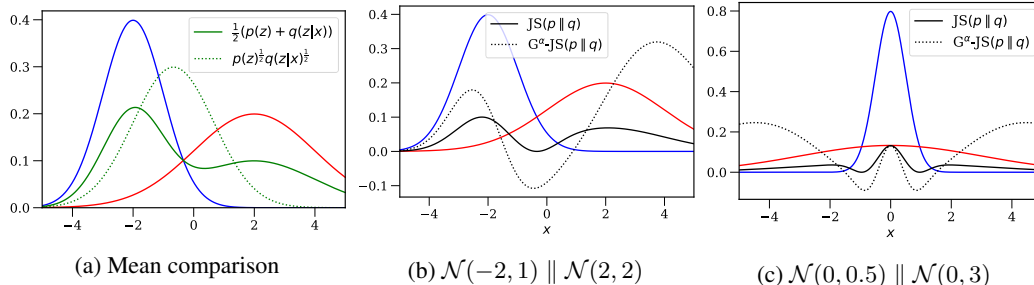

(a) Mean comparison      (b) $\mathcal{N}(-2,1) \parallel \mathcal{N}(2,2)$      (c) $\mathcal{N}(0,0.5) \parallel \mathcal{N}(0,3)$

Figure 1: Comparison of mean distributions (green) for two univariate Gaussians (red and blue), as well as comparison of arithmetic Jensen-Shannon integrand against skew-geometric Jensen-Shannon integrand with $\alpha = 0.5$ for univariate Gaussians.

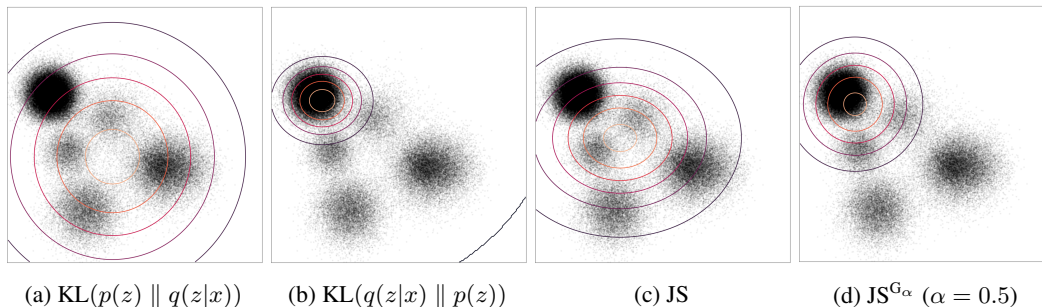

(a) $\mathrm{KL}(p(z) \parallel q(z|x))$    (b) $\mathrm{KL}(q(z|x) \parallel p(z))$    (c) JS    (d) $\mathrm{JS}^{\mathrm{G}_\alpha}$ ($\alpha = 0.5$)

Figure 2: Level sets for optimised bivariate Gaussians fit to data drawn from a mixture of Gaussians. $\mathrm{JS}^{\mathrm{G}_\alpha}$ with $\alpha$ naively set to the symmetric case $\alpha = 0.5$ (d) leads to a more reasonable intermediate distribution, when compared to (a) forward KL, (b) reverse KL and (c) Jensen-Shannon divergence. $\mathrm{JS}^{\mathrm{G}_\alpha}$ tends both towards the dominant mode and offers localised exploration.

## 3.2 Variational autoencoder benchmarks

We present quantitative evaluation results following standard experimental protocols from the literature [5, 13, 38]. In this regard, VAEs are known to have a strong capacity to reproduce images when used in conjunction with convolutional encoders and decoders. For fair comparison, we follow Higgins et al. [13] in selecting a common neural architecture across experiments[4]. Although the margin for error $\varepsilon$ in Equation (21) will vary with dataset and architecture, the point here is to standardise comparison and isolate the effect of the new divergence measure, rather than searching within architecture and hyperparameter spaces for the best performing model by some metric.

Throughout our experiments we evaluate the reconstruction loss (mean squared error) on four standard benchmark datasets: **MNIST**, $28 \times 28$ black and white images of handwritten digits [21]; **Fashion-MNIST**, $28 \times 28$ black and white images of clothing [36]; **Chairs**, $64 \times 64$ black and white images of 3D chairs [1]; **dSprites** $64 \times 64$ black and white images of 2D shapes procedurally generated from 6 ground truth independent latent factors [25].

**Influence of skew coefficient.** In Figure 3, we demonstrate several immediately useful properties of skewing our divergence away from $\alpha = 0$ or $\alpha = 1$. Firstly, intermediate skew values of $\mathrm{JS}^{\mathrm{G}_\alpha}$ do not compromise reconstruction loss and remain considerably below $\mathrm{KL}(p(z) \parallel q(z|x))$, which we find to induce the expected mode collapse across datasets. Secondly, $\mathrm{JS}^{\mathrm{G}_\alpha}$ regularisation effectively generalises to unseen data, as can be seen by the small discrepancy between train and test set evaluation. Finally, there are ranges of $\alpha$ values which produce superior reconstructions when compared to either direction of KL for identical architectures.

Furthermore, Figure 3 indicates that $\mathrm{JS}^{\mathrm{G}_\alpha}_*$ outperforms $\mathrm{KL}(q(z|x) \parallel p(z))$ for nearly all values of $\alpha$. We verify that the trend, $\mathrm{JS}^{\mathrm{G}_\alpha}$ *outperforms traditional divergences for* $\alpha < 0.3$ *and* $\mathrm{JS}^{\mathrm{G}_\alpha}_*$ *performs even better for nearly all* $\alpha$, generalises across datasets in Table 1 and Supplementary Figures 7–9. In

Figure 3c and 3d, we also include the corresponding divergence loss contributions to verify that JS$^{G\alpha}$ does not simply minimise regularisation strength in order to improve reconstruction.

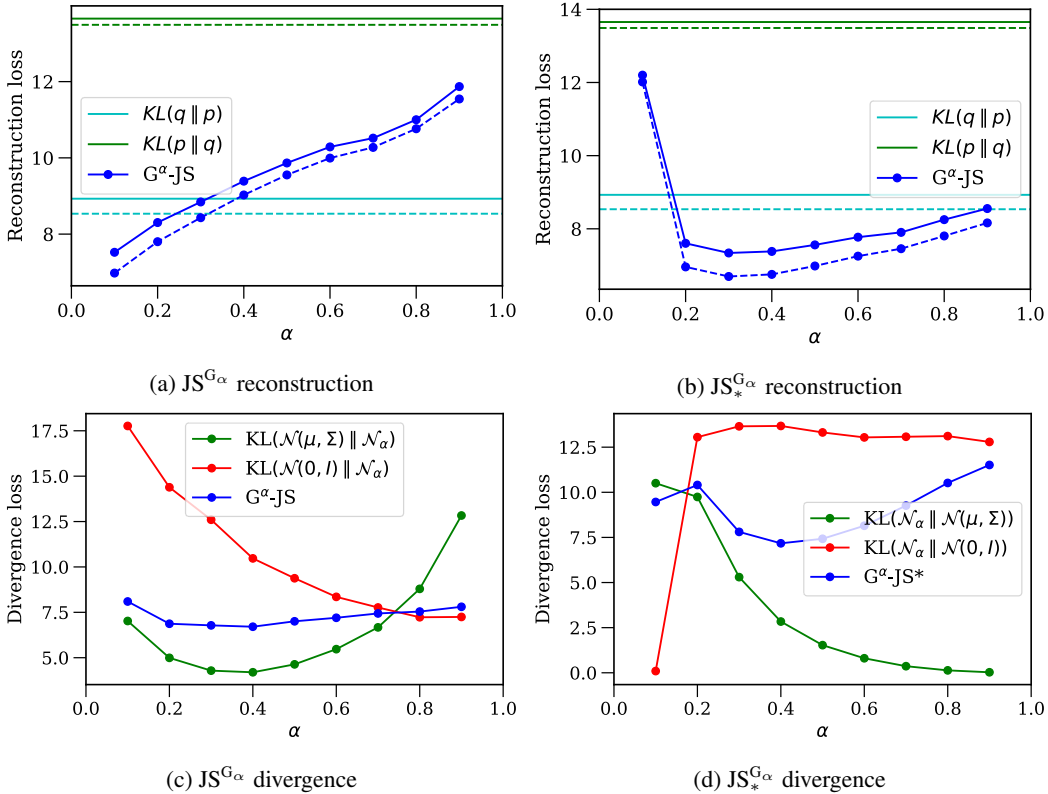

(a) JS$^{G\alpha}$ reconstruction

(b) JS$^{G\alpha}_*$ reconstruction

(c) JS$^{G\alpha}$ divergence

(d) JS$^{G\alpha}_*$ divergence

Figure 3: Reconstruction (top) and divergence (bottom) loss comparison for JS$^{G\alpha}$ (left) and JS$^{G\alpha}_*$ (right) against KL$(q(z|x) \| p(z))$ (VAE) and KL$(p(z) \| q(z|x))$ on the MNIST dataset. Throughout this work, dashed or full lines represent evaluation (sampling the mean with no variance) on the training or test sets, respectively. The comparisons performed on the other three datasets are given in Appendix B.

In Table 1, we compare the naive symmetric case JS$^{G_{0.5}}$ against the skew value with the lowest reconstruction loss (selected from $\{0.1, \ldots, 0.9\}$) for JS$^{G\alpha}$ and JS$^{G\alpha}_*$, as well as baseline regularisation terms: KL$(q(z|x) \| p(z))$, KL$(p(z) \| q(z|x))$, $\beta$-VAE (with $\beta = 4$)[5] and MMD (with $\lambda = 500$). JS$^{G\alpha}_*$ is clearly stronger than all baselines across datasets. We reinforce this point in Figure 4 where KL divergence fails to capture sharper reconstructions (such as delineating trouser legs or the heel of high-heels in the case of Fashion-MNIST) and MMD produces blurred reconstructions (we also tested $\lambda = 1000$ from [38] to no avail)[6]. More specifically, we sample each latent dimension at 10 equi-spaced points, while keeping the other 9 dimensions fixed in order to highlight the trends learnt by each dimension. As $\alpha \to 1$, the expected mode collapse occurs when approaching reverse KL across datasets, impeding reconstruction loss across more than a few modes. However, for $\alpha$ values close to 0, reverse KL images suffer from blur due to the aforementioned over-dispersion property.

**Generative capacity.** In Figure 5, we demonstrate the generative capabilities when skewing JS$^{G\alpha}$ across different $\alpha$ values. More specifically, we present the model evidence (ME) estimates for JS$^{G\alpha}$ in comparison to forward KL, reverse KL, and MMD. ME estimates are generated by Monte Carlo estimation of the marginal distribution $p_\theta(x)$ with mean and 95% confidence intervals bootstrapped from 1000 resamples of estimated batch evidence across 100 *test* set batches. We emphasise here that we are not looking for state-of-the-art results, but *relative* improvement which isolates the impact of the proposed regularisation and extends our analysis of JS$^{G\alpha}$. We see that in the case of MNIST

| Divergence | MNIST | Fashion-MNIST | dSprites | Chairs |
|---|---|---|---|---|
| $\mathrm{KL}(q(z\vert x) \parallel p(z))$ | 8.46 | 11.98 | 13.55 | 12.27 |
| $\mathrm{KL}(p(z) \parallel q(z\vert x))$ | 11.61 | 14.42 | 14.18 | 19.88 |
| $\beta$-VAE ($\beta = 4$) | 11.75 | 13.32 | 10.51 | 20.79 |
| MMD ($\lambda = 500$) | 13.19 | 11.10 | 11.87 | 18.85 |
| $\mathrm{JS}^{G_{0.5}}$ | 9.87 | 11.29 | 9.89 | 13.57 |
| $\mathrm{JS}^{G_\alpha}$ | 7.52 ($\alpha = 0.1$) | 10.04 ($\alpha = 0.2$) | 5.54 ($\alpha = 0.1$) | 11.95 ($\alpha = 0.2$) |
| $\mathrm{JS}^{G_\alpha}_*$ | **7.34** ($\alpha = 0.3$) | **9.58** ($\alpha = 0.4$) | **4.97** ($\alpha = 0.5$) | **11.64** ($\alpha = 0.4$) |

Table 1: Final model reconstruction error including optimal $\alpha$ for $\mathrm{JS}^{G_\alpha}$ and $\mathrm{JS}^{G_\alpha}_*$. The reconstruction errors for different $\alpha$ values for $\mathrm{JS}^{G_\alpha}$ and $\mathrm{JS}^{G_\alpha}_*$ are given in Appendix B

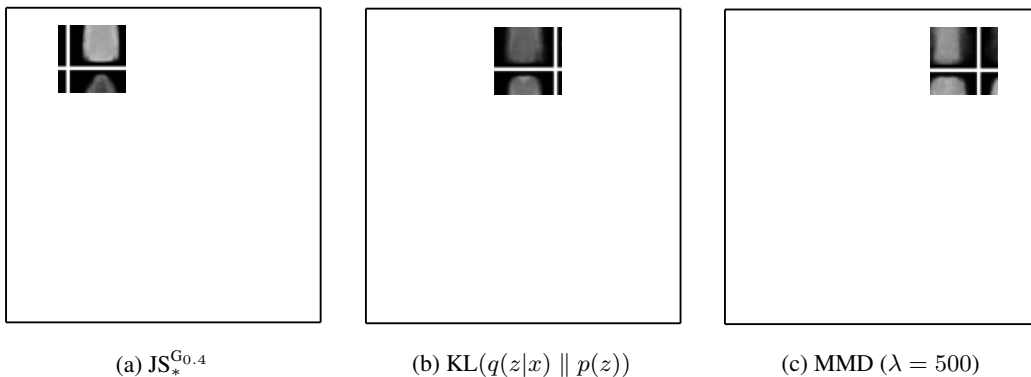

        (a) $\mathrm{JS}^{G_{0.4}}_*$          (b) $\mathrm{KL}(q(z\vert x) \parallel p(z))$          (c) MMD ($\lambda = 500$)

Figure 4: Latent space traversal for 5 of the 10 dimensions used for Fashion-MNIST. Each row represents a latent dimension and each column represents an equidistant point in the traversal.

(Figure 5a) the increased reconstructive power of $\mathrm{JS}^{G_\alpha}_*$ does come at a cost to generative performance, however, this trend is not consistent in the noisier Fashion-MNIST dataset (Figure 5b). Nevertheless, note that the reconstruction error of $\mathrm{JS}^{G_\alpha}_*$ for $\alpha > 0.8$ and $\alpha > 0.6$, in the case of MNIST and Fashion-MNIST, respectively, is still lower than the benchmarks. We also find $0.15 < \alpha < 0.4$ for $\mathrm{JS}^{G_\alpha}$ is competitive with or better than all alternatives on both datasets.

Taken all together, we make several pragmatic suggestions for selecting $\alpha$ values when using our variant of $\mathrm{JS}^{G_\alpha}$ or its dual form. Firstly, when using $\mathrm{JS}^{G_\alpha}$, lower $\alpha$ values are to be preferred, this goes some way to explaining the poor performance of the initial attempts to use $\mathrm{JS}^{G_{0.5}}$ in the literature (see Section 4). Whereas for the dual divergence, although lower $\alpha$ values ($\alpha <= 0.5$) lead to the lowest reconstruction error, higher $\alpha$ values ($\alpha > 0.6$) exhibit better generative capabilities while having lower reconstruction error than the benchmarks. Therefore, the symmetric case is a reasonably strong choice. Moreover, the plots of reconstruction loss against $\alpha$ clearly demonstrate a strong correlation between train and test set performance. This can be applied in practice, by selecting an optimal value of $\alpha$ using the training performance, circumventing the need for a separate validation set.

## 4 Related work

$\mathrm{JS}^{G_\alpha}$-VAEs build upon traditional VAEs [17, 34], with a regularisation constraint inspired by recent work on closed-form expressions for statistical divergences [30, 32]. $\mathrm{JS}^{G_\alpha}$-VAEs, offer simpler and more intuitive regularisation by skewing the intermediate distribution, allowing interpolation between forward and reverse KL divergence, and therefore combating the issue of posterior collapse [24]. In this regard, our work is related to approaches that address this issue through KL annealing during training [5, 14]. In a more general sense, this work is also related to other approaches that utilise

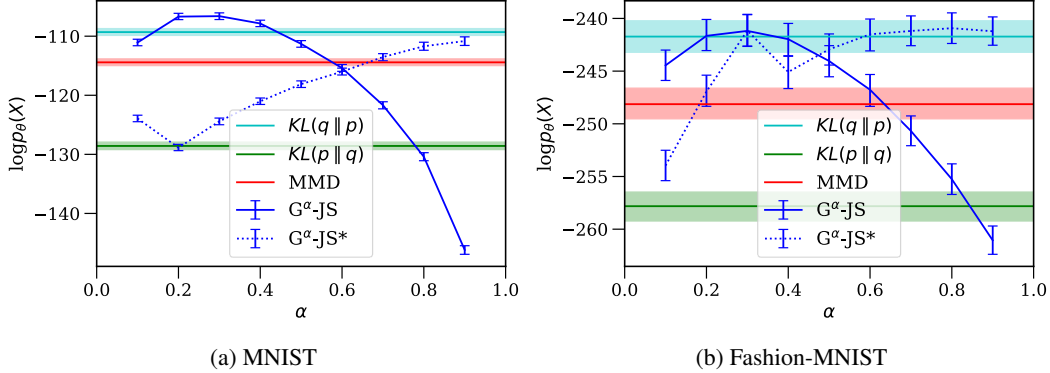

Figure 5: Estimated log model evidence and confidence intervals for $\mathrm{JS}^{\mathrm{G}_\alpha}$ and $\mathrm{JS}^{\mathrm{G}_\alpha}_*$ across different $\alpha$ values compared against $\mathrm{KL}(q(z|x) \parallel p(z))$, $\mathrm{KL}(p(z) \parallel q(z|x))$ and MMD on the (a) MNIST and (b) Fashion-MNIST datasets.

various statistical divergences and distances for latent space regularisation as an alternative to the conventional KL divergence [8, 12, 22, 37, 38].

Since its recent introduction, [2] used $\mathrm{JS}^{\mathrm{G}_{0.5}}$ as a plug-and-play replacement for JS divergence with little success, while [35] used $\mathrm{JS}^{\mathrm{G}_{0.5}}$ to decompose and estimate a multimodal ELBO loss. In contrast to these papers, we do not overlook the potential of $\mathrm{JS}^{\mathrm{G}_\alpha}$. We reverse the intermediate distribution parameterisation, allowing a principled interpolation of forward and reverse KL, we simplify the subsequent closed-form loss to that needed for VAEs, and we demonstrate improved empirical performance against several baselines (application, rather than the theory of [31]). Our more natural parameterisation and pragmatic advice on how to properly use the skew parameter $\alpha$ ultimately lead to better image reconstruction. We are not aware of any prior work exploring the dual form $\mathrm{JS}^{\mathrm{G}_\alpha}_*$.

## 5 Conclusion

Prior work assumed that no tractable interpolation existed between forward and reverse KL for multivariate Gaussians. We have overcome this with our variant of $\mathrm{JS}^{\mathrm{G}_\alpha}$, before translating it to the variational learning setting with $\mathrm{JS}^{\mathrm{G}_\alpha}$-VAE. The benefits of our variant of $\mathrm{JS}^{\mathrm{G}_\alpha}$ include symmetry (at $\alpha = 0.5$) and having closed-form expression. Alongside this, we have demonstrated that the advantages of its role in VAEs include quantitatively and qualitatively better reconstructions than several baselines. Although we accept that use of "vanilla" VAEs may not out-compete some of the leading flow and GAN based architectures, we believe our regularisation mechanism addresses the trade-off between zero-avoidance and zero-forcing in latent space, which goes some way to bridge this gap while being intuitive in both divergence and distribution space. Our experiments demonstrate that the flexibility accorded to VAEs by skewing $\mathrm{JS}^{\mathrm{G}_\alpha}$ is worth considering across a broad range of applications.

## Broader Impact

For the statistics community, our introduction of the alternative $\text{JS}^{\text{G}_{\alpha'}}$ and $\text{JS}^{\text{G}_{\alpha'}}_*$, rather than the "original" $\text{JS}^{\text{G}_\alpha}$ and $\text{JS}^{\text{G}_\alpha}_*$, immediately presents a benefit as a more intuitive interpolation through divergence and distribution space. As we have shown the benefits of such an interpolation on the task of image reconstruction, the first impact of our model lies in better image compression and generation from latent samples. However, in a more general setting, VAEs present multiple impactful opportunities.

Applications include compression (of any data type), generation of new samples in fields with data paucity, as well as extraction of underlying relationships. As our exploration of the $\text{JS}^{\text{G}_\alpha}$ family of VAEs has improved performance, after translation to data types with other structures, our VAE could be used for all of these applications. Our experiments also indicate strong regions for the skew parameter $\alpha$ which could be used as a standard regularisation mechanism across variational learning.

In settings with sensitive data, all of these applications bear some risks. As VAEs provide a form of *lossy* compression, in healthcare and social settings there is the risk of misrepresenting personal information in latent space. In areas of data paucity, without additional constraints, VAEs may generate samples which are unrealistic and severely bias any downstream training. Finally, when using VAEs in science, to extract underlying associations, it remains important to analyse the true meaning of any independent components extracted, rather than taking these rules at face value.

## Acknowledgments and Disclosure of Funding

We thank Cristian Bodnar, Cătălina Cangea, Ben Day, Felix Opolka, Emma Rocheteau, Ramon Viñas Torne and Duo Wang from the Department of Computer Science and Technology, University of Cambridge, for their helpful comments. We would like to also thank the reviewers for their constructive feedback and efforts towards improving our paper. We acknowledge the support of The Mark Foundation for Cancer Research and Cancer Research UK Cambridge Centre [C9685/A25177] for N.S. The authors declare no competing interests.

## Footnotes

[2]Code is available at: https://github.com/jacobdeasy/geometric-js

[3]Details on the influence of $\lambda$ on the reconstructive performance of VAEs, with respect to $\mathrm{JS}^{\mathrm{G}_\alpha}$ and $\mathrm{JS}^{\mathrm{G}_\alpha}_*$, are given in Appendix E

[4]The specific model details are given in Appendix C

[5]Details on the performance of $\beta$-VAEs for varying $\beta$ is given in Appendix F

[6]Additional qualitative analyses are prestened in Appendix G

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
