[Supplementary Material]

# A Proofs

## A.1 Proof of proposition 1

*Proof.* We first present the more general case of distributions $p$ and $q$ permitting a geometric mean distribution (e.g. $p$ and $q$ members of the exponential family), as we believe this more general case to be of note.

$$\text{JS}^{G_{\alpha'}} = (1-\alpha)\text{KL}\left(p \parallel G_{\alpha'}(p,q)\right) + \alpha\text{KL}\left(q \parallel G_{\alpha'}(p,q)\right) \tag{27}$$

$$= (1-\alpha)\text{KL}\left(p \parallel p^\alpha q^{1-\alpha}\right) + \alpha\text{KL}\left(q \parallel p^\alpha q^{1-\alpha}\right) \tag{28}$$

$$= (1-\alpha)\int_x p\log\left[\frac{p}{p^\alpha q^{1-\alpha}}\right]dx + \alpha\int_x q\log\left[\frac{q}{p^\alpha q^{1-\alpha}}\right]dx \tag{29}$$

$$= (1-\alpha)^2\int_x p\log\left[\frac{p}{q}\right]dx + \alpha^2\int_x q\log\left[\frac{q}{p}\right]dx \tag{30}$$

$$= (1-\alpha)^2\text{KL}(p \parallel q) + \alpha^2\text{KL}(q \parallel p) \tag{31}$$

Therefore, the respective cases disappear in the limits $\alpha \to 0$ and $\alpha \to 1$ and for $\text{JS}^{G_{\alpha'}}$ we have, in fact, recovered an equivalence between linear scaling in distribution space and quadratic scaling in the space of divergences.

The dual case $\text{JS}_*^{G_{\alpha'}}$ does not simplify in the same way because the geometric mean term lies outside of the logarithm. However, instead we have

$$\text{JS}_*^{G_{\alpha'}} = (1-\alpha)\text{KL}\left(G_{\alpha'}(p,q) \parallel p\right) + \alpha\text{KL}\left(G_{\alpha'}(p,q) \parallel q\right) \tag{32}$$

$$= (1-\alpha)\text{KL}\left(p^\alpha q^{1-\alpha} \parallel p\right) + \alpha\text{KL}\left(p^\alpha q^{1-\alpha} \parallel q\right) \tag{33}$$

$$= (1-\alpha)\int_x p^\alpha q^{1-\alpha}\log\left[\frac{p^\alpha q^{1-\alpha}}{p}\right]dx + \alpha\int_x p^\alpha q^{1-\alpha}\log\left[\frac{p^\alpha q^{1-\alpha}}{q}\right]dx \tag{34}$$

$$= (1-\alpha)^2\int_x p^\alpha q^{1-\alpha}\log\left[\frac{q}{p}\right]dx + \alpha^2\int_x p^\alpha q^{1-\alpha}\log\left[\frac{p}{q}\right]dx. \tag{35}$$

The final step is to recognise the two limits

$$\lim_{\alpha\to 0}\left[p^\alpha q^{1-\alpha}\right] = q \qquad\qquad \lim_{\alpha\to 1}\left[p^\alpha q^{1-\alpha}\right] = p, \tag{36}$$

mean that we recover

$$\lim_{\alpha\to 0}\left[\text{JS}_*^{G_{\alpha'}}\right] = \text{KL}\left(\mathcal{N}_2 \parallel \mathcal{N}_1\right) \qquad\qquad \lim_{\alpha\to 1}\left[\text{JS}_*^{G_{\alpha'}}\right] = \text{KL}\left(\mathcal{N}_1 \parallel \mathcal{N}_2\right). \tag{37}$$

$\square$

Overall, although the limiting cases are reversed between $\text{JS}^{G_{\alpha'}}$ and $\text{JS}_*^{G_{\alpha'}}$, we note that the approach to either limiting case is distinct and comes with its own benefits through the weighting (non-logarithmic) term used in the integrand.

## A.2 Proof of proposition 2

We choose to prove proposition 1 via reduction of the form in Equation (9), although we note it is also reasonable to simply follow through the weighted sum in Equation (8).

*Proof.* After defining $\Sigma_{ii} = \sigma_i^2$, $(\Sigma_\alpha)_{ii} = \sigma_{\alpha,i}^2$ and $(\mu_\alpha)_i = \mu_{\alpha,i}$, it is apparent $\Sigma_2 = I$ gives

$$\sigma_{\alpha,i}^2 = \frac{1}{((1-\alpha)\sigma_i^2 + \alpha)}, \tag{38}$$

and $\mu_2 = 0$ (the zero vector) gives

$$\mu_{\alpha,i} = \sigma_{\alpha,i}^2\left((1-\alpha)\frac{\mu_i}{\sigma_i^2}\right) \tag{39}$$

We can then reduce Equation (9) using diagonal matrix properties

$$\mathrm{JS}^{\mathrm{G}_\alpha}\left(\mathcal{N}_1 \parallel \mathcal{N}_2\right) = \frac{1}{2}\left(\sum_{i=1}^{n}\frac{1}{\sigma_{\alpha,i}^2}\left((1-\alpha)\sigma_i^2+\alpha\right)+\log\left[\frac{\prod\limits_{i=1}^{n}\sigma_{\alpha,i}^2}{\prod\limits_{i=1}^{n}(\sigma_i^2)^{1-\alpha}}\right]\right. \tag{40}$$

$$\left.+\frac{(1-\alpha)(\mu_{\alpha,i}-\mu_i)^2}{\sigma_{\alpha,i}^2}+\frac{\alpha\mu_{\alpha,i}^2}{\sigma_{\alpha,i}^2}-n\right), \tag{41}$$

and application of log laws recovers Equation (23).

The proof of the dual form in Equation (25) is carried out similarly.  □

## B  Additional training and evaluation information

| Divergence | MNIST | Fashion-MNIST | dSprites | Chairs |
|---|---|---|---|---|
| $\mathrm{KL}(q(z\|x) \parallel p(z))$ | 8.46 | 11.98 | 13.55 | 12.27 |
| $\mathrm{KL}(p(z) \parallel q(z\|x))$ | 11.61 | 14.42 | 14.18 | 19.88 |
| $\beta$-VAE ($\beta = 4$) | 11.75 | 13.32 | 10.51 | 20.79 |
| $\beta$-VAE ($\beta = 0.25$) | 8.09 | *9.07* | 10.39 | 14.09 |
| MMD ($\lambda = 500$) | 13.19 | 11.10 | 11.87 | 18.85 |
| $\mathrm{JS}^{\mathrm{G}_{0.1}}$ | **7.52** | 10.04 | **6.63** | 12.62 |
| $\mathrm{JS}^{\mathrm{G}_{0.2}}$ | 8.30 | **10.04** | 7.50 | **11.95** |
| $\mathrm{JS}^{\mathrm{G}_{0.3}}$ | 8.84 | 10.50 | 8.56 | 12.40 |
| $\mathrm{JS}^{\mathrm{G}_{0.4}}$ | 9.39 | 10.93 | 9.16 | 12.96 |
| $\mathrm{JS}^{\mathrm{G}_{0.5}}$ | 9.87 | 11.29 | 9.89 | 13.57 |
| $\mathrm{JS}^{\mathrm{G}_{0.6}}$ | 10.28 | 11.72 | 10.38 | 14.15 |
| $\mathrm{JS}^{\mathrm{G}_{0.7}}$ | 10.51 | 12.09 | 10.80 | 14.68 |
| $\mathrm{JS}^{\mathrm{G}_{0.8}}$ | 11.00 | 12.44 | 11.40 | 15.48 |
| $\mathrm{JS}^{\mathrm{G}_{0.9}}$ | 11.87 | 13.21 | 12.05 | 16.27 |
| $\mathrm{JS}_*^{\mathrm{G}_{0.1}}$ | 12.20 | 13.52 | 5.54 | 15.53 |
| $\mathrm{JS}_*^{\mathrm{G}_{0.2}}$ | 7.60 | 10.90 | 5.18 | 13.06 |
| $\mathrm{JS}_*^{\mathrm{G}_{0.3}}$ | **7.34** | 10.51 | 5.06 | 12.09 |
| $\mathrm{JS}_*^{\mathrm{G}_{0.4}}$ | 7.38 | **9.58** | 5.17 | **11.64** |
| $\mathrm{JS}_*^{\mathrm{G}_{0.5}}$ | 7.56 | 9.80 | **4.97** | 11.75 |
| $\mathrm{JS}_*^{\mathrm{G}_{0.6}}$ | 7.77 | 10.01 | 5.30 | 12.07 |
| $\mathrm{JS}_*^{\mathrm{G}_{0.7}}$ | 7.90 | 10.34 | 5.23 | 12.53 |
| $\mathrm{JS}_*^{\mathrm{G}_{0.8}}$ | 8.25 | 10.84 | 5.42 | 13.11 |
| $\mathrm{JS}_*^{\mathrm{G}_{0.9}}$ | 8.55 | 11.40 | 5.74 | 13.52 |

Table 2: Final model reconstruction error for different $\alpha$ values for $\mathrm{JS}^{\mathrm{G}_\alpha}$ and $\mathrm{JS}_*^{\mathrm{G}_\alpha}$.

(a) $\text{JS}^{\text{G}_\alpha}$ reconstruction.

(b) $\text{JS}^{\text{G}_\alpha}_*$ reconstruction.

(c) $\text{JS}^{\text{G}_\alpha}$ divergence.

(d) $\text{JS}^{\text{G}_\alpha}_*$ divergence.

Figure 6: Breakdown of final model loss components on the MNIST dataset.

(a) $\text{JS}^{\text{G}_\alpha}$ reconstruction.

(b) $\text{JS}^{\text{G}_\alpha}_*$ reconstruction.

(c) $\text{JS}^{\text{G}_\alpha}$ divergence.

(d) $\text{JS}^{\text{G}_\alpha}_*$ divergence.

Figure 7: Breakdown of final model loss on the Fashion-MNIST dataset.

(a) $\mathrm{JS}^{\mathrm{G}_\alpha}$ reconstruction.

(b) $\mathrm{JS}^{\mathrm{G}_\alpha}_*$ reconstruction.

(c) $\mathrm{JS}^{\mathrm{G}_\alpha}$ divergence.

(d) $\mathrm{JS}^{\mathrm{G}_\alpha}_*$ divergence.

Figure 8: Breakdown of final model loss components on the dSprites dataset.

(a) $\mathrm{JS}^{\mathrm{G}_\alpha}$ reconstruction.

(b) $\mathrm{JS}^{\mathrm{G}_\alpha}_*$ reconstruction.

(c) $\mathrm{JS}^{\mathrm{G}_\alpha}$ divergence.

(d) $\mathrm{JS}^{\mathrm{G}_\alpha}_*$ divergence.

Figure 9: Breakdown of final model loss components on the Chairs dataset.

# C  Model details

We use the architectures specified in Table 3 throughout experiments. We pad 28x28x1 images to 32x32x1 with zeros as we found resizing images negatively affected performance. We use a learning rate of 1e-4 throughout and use batch size 64 and 256 for the two MNIST variants and the other datasets respectively. Where not specified (e.g. momentum coefficients in Adam [16]), we use the default values from PyTorch [33]. The only architectural change we make between datasets is an additional convolutional (and transpose convolutional) layer for encoding (and decoding) when inputs are 64x64x1 instead of 32x32x1. We train dSprites for 30 epochs and all other datasets for 100 epochs.

| Dataset | Stage | Architecture |
|---|---|---|
| MNIST | Input | 28x28x1 zero padded to 32x32x1. |
| | Encoder | Repeat Conv 32x4x4 for 3 layers (stride 2, padding 1). FC 256, FC 256. ReLU activation. |
| | Latents | 10. |
| | Decoder | FC 256, FC 256, Repeat Deconv 32x4x4 for 3 layers (stride 2, padding 1). ReLU activation, Sigmoid. MSE. |
| Fashion-MNIST | Input | 28x28x1 zero padded to 32x32x1. |
| | Encoder | Repeat Conv 32x4x4 for 3 layers (stride 2, padding 1). FC 256, FC 256. ReLU activation. |
| | Latents | 10. |
| | Decoder | FC 256, FC 256, Repeat Deconv 32x4x4 for 3 layers (stride 2, padding 1). ReLU activation, Sigmoid. Bernoulli. |
| dSprites | Input | 64x64x1. |
| | Encoder | Repeat Conv 32x4x4 for 4 layers (stride 2, padding 1). FC 256, FC 256. ReLU activation. |
| | Latents | 10. |
| | Decoder | FC 256, FC 256, Repeat Deconv 32x4x4 for 4 layers (stride 2, padding 1). ReLU activation, Sigmoid. Bernoulli. |
| Chairs | Input | 64x64x1. |
| | Encoder | Repeat Conv 32x4x4 for 4 layers (stride 2, padding 1). FC 256, FC 256. ReLU activation. |
| | Latents | 32. |
| | Decoder | FC 256, FC 256, Repeat Deconv 32x4x4 for 4 layers (stride 2, padding 1). ReLU activation, Sigmoid. Bernoulli. |

Table 3: Detail of model architectures.

# D $\mathbf{JS^{G_{\alpha'}}}$ vs. $\mathbf{JS^{G_{\alpha}}}$

(a) $JS^{G_\alpha}$.

(b) $JS^{G_\alpha}_*$.

Figure 10: Comparison of the original $JS^{G_\alpha}$ and our variant, $JS^{G_{\alpha'}}$, on the MNIST dataset.

(a) $JS^{G_\alpha}$.

(b) $JS^{G_\alpha}_*$.

Figure 11: Comparison of the original $JS^{G_\alpha}$ and our variant, $JS^{G_{\alpha'}}$, on the Fashion-MNIST dataset.

# E Influence of the $\lambda$ parameter on the performance of $\mathrm{JS}^{G_\alpha}$-VAEs and $\mathrm{JS}^{G_\alpha}_*$-VAEs

(a) $\mathrm{JS}^{G_\alpha}$.  (b) $\mathrm{JS}^{G_\alpha}_*$.

Figure 12: Comparison of the reconstruction loss of $\mathrm{JS}^{G_\alpha}$-VAEs and $\mathrm{JS}^{G_\alpha}_*$-VAEs for different values of $\lambda$, on the MNIST dataset.

# F Performance of $\beta$-VAEs for varying $\beta$

Figure 13: Comparison of the reconstruction loss of $\beta$-VAEs for different values of $\beta$, on the MNIST dataset.

# G  Latent samples

(a) $\mathrm{JS}_*^{\mathrm{G}_{0.1}}$.

(b) $\mathrm{JS}_*^{\mathrm{G}_{0.4}}$.

(c) $\mathrm{JS}^{\mathrm{G}_{0.9}}$.

Figure 14: Latent space traversal of Fashion-MNIST for different skew values of $\mathrm{JS}_*^{\mathrm{G}_\alpha}$.

(a) $\mathrm{JS}_*^{\mathrm{G}_{0.1}}$

(b) $\mathrm{JS}_*^{\mathrm{G}_{0.5}}$

(c) $\mathrm{JS}_*^{\mathrm{G}_{0.9}}$

(d) $\mathrm{KL}(q(z|x) \parallel p(z))$

Figure 15: Latent space traversal dSprites for different skew values and KL divergence.

(a) $\text{JS}_*^{\text{G}_{0.4}}$.

(b) $\text{KL}(q(z|x) \parallel p(z))$

Figure 16: Latent space traversal for the Chairs dataset (32 latent dimensions).