[Reviews · NeurIPS 2020]

Review 1

Summary and Contributions: This paper makes 3 contributions. First, it explores the properties of the skew geometric-Jensen-Shannon divergence (JS^G_\alpha) with respect to classic Kullback-Leibler (KL) and Jensen-Shannon (JS) divergence. Second, the authors modify the JS^G_\alpha and reverse the skew parameter in the geometric distribution in order to obtain a divergence with more intuitive properties in limit cases. Finally, the authors use the JS^G_\alpha in the Variational Autoencoders (VAE) setting in replacement of the Kullback-Leibler (KL) divergence and perform experiments on several benchmark datasets to show the relevance of their method.

Strengths: Strengths: * The author provide an extensive analysis of the property of the divergence employed, which makes their claim stronger and grounded. * The method is theoretically grounded and good empirical evaluation is performed. * I believe the method to be novel (in its use of the dual, the alternative divergence and the selection of the parameter \alpha) and of good significance and relevance to the community.

Weaknesses: Weaknesses: * The reconstruction are not of great quality, which is a general problem when using VAE-type of methods.

Correctness: I believe the method, the claims and empirical methodology to be thorough and correct.

Clarity: The paper is very well written, easy to read and understand.

Relation to Prior Work: Yes

Reproducibility: Yes

Additional Feedback: I really liked the simplicity of the method and the clarity of the paper. I thank the authors for explaining very well their method, and for the preliminary analysis of the behavior of JS^G_alpha with respect to different values of \alpha and the KL divergence. Suggestions for improvements and questions : * Figure 4 is too small to conclude anything (apart from the poor performances of MMD) * Small typos: equation (19) N_1 and N_2 are inversed in the dual divergence limit case (if I am correct). * Disentanglement is a key challenge in ML where VAE-based methods are often employed. Could the skew geometric JS divergence be applied there ? * Would the authors have ideas on how to improve reconstruction quality ?


Review 2

Summary and Contributions: Post author rebuttal: Having read the reviews and author's rebuttal, I am also inclined to keep my score as it is. The paper is clearly written and the extra experiments provided in the rebuttal adds confidence to the paper. The paper suggests a new regularization in the latent space for VAE, using skewed geometric-Jensen-Shannon divergences. The proposed divergence leads to an intuitive interpolation between forward and reverse KL that trades off zero-avoidance and zero-forcing behaviour. The additional hyperparameter introduced which can be easily interpreted in latent space. Experiments showed promising results on a couple of datasets.

Strengths: The divergence proposed is novel and has an intuitive interpretation. The closed analytical form makes it easy to use in practice and the experiments showed promising results compared with other VAE variants. The method is theoretically sound and is related to the NeurIPS audience.

Weaknesses: The experiments can be more extensive. The tightness of the ELBO with this new regularization and the probabilistic interpretation of it can be extended.

Correctness: Yes, the authors provided proofs for the theorem and experimental results.

Clarity: Yes, the paper is clearly written.

Relation to Prior Work: Yes, related work are discussed in section 1.

Reproducibility: Yes

Additional Feedback: How was the beta parameter selected for beta-VAE? How does it compare with Wasserstein VAE? Would be interesting to see the limiting case of \alpha in Figure 4. what's the complexity of the algorithm?


Review 3

Summary and Contributions: The paper proposes an alternative regularization divergence for variational autoencoders (VAEs). Specifically, the paper proposes to use the skew geometric Jensen-Shannon divergence in lieu of the KL divergence in the variational objective of VAEs. The paper’s main contributions are theoretical results that show that the skew geometric Jensen-Shannon divergence allows for interpolation between forward and reverse KL divergences and that the skew geometric Jensen-Shannon divergence from a diagonal to a standard Gaussian distribution, which are commonly used in VAEs, can be obtained as a simple, analytical expression. Moreover, the authors provide an empirical investigation that provides an intuition for the proposed divergence and demonstrates that the skew geometric Jensen-Shannon divergence can lead to improved performance in VAEs compared to other divergence regularization terms.

Strengths: ++ Preface ++ I understand that reviews that claim that a method is not sufficiently novel or significant are often subjective and are difficult for authors to rebut. To make my review easier to engage with, I’m offering the following criteria along which I assess “significance” of a paper: (*i*) Does the paper offer a novel, non-obvious theoretical insight in the form of a proof or derivation? (*ii*) Does the paper propose a new method that leverages novel, non-trivial theoretical, conceptual, and/or empirical insights? (*iii*) Does the paper present believable/verifiable results that solve a previously unsolved problem or that significantly improve upon existing methods. I will touch on these three criteria in my comments below and mark my comment accordingly. ++++++++++++++++++++++++++++++++++++++++++++++++++++++++++++++ ++ Relevance ++ The effect of different regularization terms in VAEs in an active area of research. Specifically, recent work on the role of regularization in VAEs has tied regularization to interpretability of VAEs via disentanglement. This paper provides a novel perspective on regularization in VAEs by considering a class of divergences previously discarded as not suitable for training VAEs. As such, I believe that the paper is relevant to the NeurIPS community. ++++++++++++++++++++++++++++++++++++++++++++++++++++++++++++++ ++ Significance and novelty of contribution ++ (*i*) Does the paper offer a novel, non-obvious theoretical insight in the form of a proof or derivation? The paper builds on prior work on the Jensen-Shannon family of divergences and shows that under a certain parameterization, the skew geometric Jensen-Shannon distribution allows for interpolation between forward and reverse KL. While I am not intimately familiar with the specific literature on lambda divergences, it is my understanding that this specific insight is novel and that the application of the resulting divergence to VAEs has not been explored before. As a result, although the derivations in the appendix are simple, I believe that the key theoretical insight constitutes a significant contribution. (*ii*) Does the paper propose a new method that leverages novel, non-trivial theoretical, conceptual, and/or empirical insights? The paper uses the reparameterized skew geometric Jensen-Shannon divergence as regularization in VAEs. To do so, it follows the same derivations as Higgins et al. (https://openreview.net/forum?id=Sy2fzU9gl and derives a novel variational objective from a constrained optimization problem with the constraint that the skew geometric Jensen-Shannon divergence be under a certain threshold. The paper points out that the beta-VAE objective is a special case of their objective. To make this objective more tractable, the paper presents an analytic expression of the skew geometric Jensen-shannon divergence between a Gaussian distribution with diagonal covariance and a standard Gaussian distribution, as commonly used in VAEs. I believe that the derivation of the reparameterized skew geometric Jensen-Shannon divergence, the derivation of the analytical expression for amortized inference in VAEs, and the conceptual explanations and intuitions provided in Section 3 constitute an insightful contribution. ++++++++++++++++++++++++++++++++++++++++++++++++++++++++++++++ ++ Soundness of theoretical claims ++ I checked the derivations in the appendix and did not find any errors. ++++++++++++++++++++++++++++++++++++++++++++++++++++++++++++++ ++ Quality and soundness of empirical claims ++ The paper claims that using the reparameterized skew geometric Jensen-Shannon divergence leads to better image reconstructions and better image generation and back up their claim with a set of simple experiments where they compare their method to VAEs with alternative regularizers and show that subject to a well-chosen skewness parameters, the proposed method outperforms VAEs with KL regularization. The authors also provide nicely documented code including several notebooks that reproduce their results. ++++++++++++++++++++++++++++++++++++++++++++++++++++++++++++++

Weaknesses: ++++++++++++++++++++++++++++++++++++++++++++++++++++++++++++++ ++ Relevance ++ See under “Strengths” above. ++++++++++++++++++++++++++++++++++++++++++++++++++++++++++++++ ++ Significance and novelty of contribution ++ I want to highlight again that I am not intimately familiar with the literature on lambda divergences cited in the paper and, as a result, cannot provide an independent assessment of the correctness of the authors’ claim regarding the extent to which the reparameterization of the skew geometric Jensen-Shannon divergence constitutes a significant contribution. That being said, I believe that even if the reparameterization has been explored in the literature on lambda divergence, the connection and application to VAEs by itself constitutes a veritable contribution. ++++++++++++++++++++++++++++++++++++++++++++++++++++++++++++++ ++ Soundness of theoretical claims ++ See under “Correctness” below. ++++++++++++++++++++++++++++++++++++++++++++++++++++++++++++++ ++ Quality and soundness of empirical claims ++ See under “Strengths” above. ++++++++++++++++++++++++++++++++++++++++++++++++++++++++++++++

Correctness: I checked the proofs/derivations in the appendix and did not find any errors. In line 132 in the paper, it says that the beta-VAE objective is recovered for alpha = 1, but based on Equations (18) and (19), the beta-VAE objective should be recovered for alpha = 0. It is possible that there is some confusion between alpha and alpha prime, but I would appreciate it if the authors could briefly address if this is a mistake or not.

Clarity: ++++++++++++++++++++++++++++++++++++++++++++++++++++++++++++++ ++ Strengths ++ The organization of the paper is good, the figures are insightful and look good, and the derivations are easy to follow. ++++++++++++++++++++++++++++++++++++++++++++++++++++++++++++++ ++ Weaknesses ++ The presentation of the paper in terms of the quality of the writing, figure captions, and the abstract needs significant improvement. - The abstract is wordy, presents unnecessary information (e.g., the dimensionality of the data points and the latent space), and does not motivate the problem very well. I recommend significantly rewriting the abstract with outside feedback. Should the paper get accepted, I would recommend to the authors that they request that they be allowed to significantly rewrite the abstract. This may be possible if it was explicitly requested by reviewers and/or the area chair. - The first paragraph of Section 3.1 is not very clear and leaves me wondering what this section is supposed to communicate. In any case, it leaves me wondering why the plots in Figure (1) show that the skew geometric Jensen-Shannon divergence is better than the alternatives. - The captions of most figures contain errors or do not provide sufficient information for interpreting the plots. The captions of (a), (b), and (c) in Figure (1) are confusing and barely tell me anything. What does “mean comparison” mean here? What does \mathcal{N}(-2, 1) || \mathcal{N}(2, 2) mean? The caption of Figure 1 is also unclear. Similarly, what does “JS.” in Figure (2c) mean? The caption of Figure (2d) is incomplete. - There are several language and style errors in the paper, for example, it should be “the KL divergence” and not “KL divergence”, i.e., in the majority of cases KL or JS are used in the paper, it should say “the KL divergence” or “the JS divergence”. Currently, the definite article (i.e., “the”) and the term “divergence” are missing throughout, see for example, lines 54, 60, 65, 93, and numerous other places. Other errors: - Line 148: “we have outlined JS” -> “we have discussed the JD divergence” - Line 172: “reproduce” -> “reconstruct” - Line 775: “with dataset” -> “with the dataset” - Line 176: “standardise comparison” -> “standardise the comparison” - Line 191: Missing word in “We verify that the trend, JS…” - Line 206: “We draw several pragmatic suggestions” -> “We make several pragmatic suggestions” - Line 208: “to explaining poor performance” -> “in explaining the poor performance” - Line 231: “and closed-form” -> “and the existence of a closed-form expression” - Line 232: “we have demonstrated that the advantages of its role in VAEs include…” -> “we have demonstrated the advantages of the JS divergence in VAEs, including…” - Line 235: “which goes some way to bridge this gap” -> “which goes some way in bridging this gap”

Relation to Prior Work: The related work section is very short and does not offer much insight to the reader. Lines 222-226 are helpful in placing the proposed objective in the context of prior work, but lines 214-221 do not actually provide much context. I believe that the authors could relate their method more clearly to prior work on disentanglement and interpretability in VAEs. Additionally, a more thorough comparison to prior work on lambda divergences would help the reader better understand (and potentially also to better appreciate) the significance of the reparameterization.

Reproducibility: Yes

Additional Feedback: I strongly recommend to the authors to improve the presentation of the paper and fix the issues listed above. Please do not feel obligated to respond to any minor concerns I expressed in this review. I would appreciate it if the authors could address my concern under “Correctness”. I would also appreciate a better explanation as to why the authors think Figure (1) shows that the skew geometric Jensen-Shannon divergence is (objectively?) better than the alternatives. Additionally, I would like the authors to explain to what extent this work goes beyond Nielsen (2019, https://www.mdpi.com/1099-4300/21/5/485), which (contrary to what's being asserted in the paper under review) proposes the dual of the skew geometric Jensen-Shannon and also provides analytical expressions for the divergence. *************************** POST-REBUTTAL UPDATE *************************** I thank the authors for addressing my questions and concerns. My score of "6: Marginally above the acceptance threshold." remains unchanged: (1) The contribution of the paper is relevant, but as outlined in my review, the paper's main contribution of reversing the parameterization of the divergence, which (along with its dual) was previously derived in Nielsen (2019), and applying it to VAEs seems like a minor contribution to me. (2) Additionally, as the paper needs significant editing before being publishable. I hope the authors will follow Reviewer 4's and my suggestions for improving the paper.


Review 4

Summary and Contributions: This paper introduces the skew geometric Jensen-Shannon Variational Autoencoder (JSG-VAE). It replaces the KL(q(z|x) || p(z)) term in the standard ELBO expression with an *adjusted* skew geometric Jensen-Shannon divergence (aJSG), which is the main contribution of the paper. Similarly to the skew geometric Jensen-Shannon divergence, the adjusted version is controlled via a single hyperparameter. The paper shows that the proposed method includes other methods for training VAEs, including the standard ELBO and beta-VAEs (Higgins et al., 2017), as a limiting case of the value of the hyperparameter. Various desirable properties of the aJSG is then discussed both theoretically and experimentally. The JSG-VAE is finally compared to other baseline VAEs on MNIST, Fashion-MNIST, dSprites, and Chairs, where the JSG-VAE is shown to outperform all compared methods, by considering the reconstruction error as a metric.

Strengths: - The aJSG is theoretically well justified and ties in well with prior work. - Although the introduced adjustment to the skew geometric Jensen-Shannon divergence (vanilla JSG) itself is very simple, the behavioral change it brings to the vanilla JSG is highly significant. Specifically, it is this change which makes possible a natural interpolation between the forward and backward KL divergence by adjusting the hyperparameter. Not only is this a desirable practical feature of the proposed method, but it brings a principled way for controlling the divergence term of the VAE objective with known and well-defined limits with respect to the hyperparameter. The vanilla JSG allows for no such interpolation. - Like the KL divergence, the proposed aJSG divergence also allows for closed form solutions when considering distributions in the same exponential family (Frank Nielsen, 2019). Note that this is also true for the standard KL divergence. - The experimental results are convincing and the proposed method is properly compared against various baselines (i.e. considering different divergences to regularize the reconstruction error)

Weaknesses: 1) While the experiments show superior performance of the proposed method with respect to the reconstruction error, the lack of experiments that illustrate the generative capabilities of the model is the main weakness of the paper. To be more specific: i) When compared to using the classic ELBO: in this case it is well known that if the learned posterior matches the true posterior, the subsequent update will increase the model evidence p_theta(x) ii) Changing the divergence terms in the ELBO disrupts the property in i), and depending on what that divergence is, it might be "stronger" or "weaker" in the sense that it either forces q(z|x) to be more/less "close" to p(z) compared to using the ELBO. This in turn might allow training a q(z|x) which aims solely at producing good reconstruction. That is to say, as Eq. (22) seems to illustrate, maximizing this objective is equivalent to to replacing p(z) --> int q(z|x)p(x) dx = q(z), and maximizing the model evidence under this other prior (q(z)) may lead to poor generation when sampling z~p(z) and then x~p(x|z). ii) I do not argue that replacing the divergence term in the ELBO is certain to hurt the generative power of the method. However, including experimental results or theoretical statements that speak towards the generative power (i.e. how the objective relates to p(x)) would be informative as it could testify to either the limitation of the method or provide additional support. 2) The lack of comparison to the vanilla JSG (although Figure 12 in the appendix include a single comparison). Comparing the aJSG against vanilla JSG for all the experiments would show the importance of considering the adjusted version from an experimental point of view.

Correctness: Generally, the paper makes correct claims and presents a sound method which is properly supported by mathematical arguments. The experiments are carried out properly and support the conclusions and claims made. Two very minor things: 1) The claim made on line 193-194: I do not necessarily agree that the aJSG is obviously superior to the Jensen-Shannon (JS) divergence with respect to the zero avoiding/forcing properties. That is to say, aJSG ignores some pretty high density regions, whereas JS covers a bit more low density regions. In conclusion JS suffers from (less) zero forcing whereas aJSG suffers from (less) zero avoiding. 2) line 186-187: Both the KL(q||p) or KL(p||q) based methods show similar differences between the training curve and validation curve.

Clarity: Generally a very well written paper that is easy to follow, understand, and highly enjoyable. Well done! I did however find Figure 4 quite confusing, and had to consult the source code to fully work out what I was looking at. The paper leaves too much for the reader to interpret regarding Figureu 4. The paper should guide the reader more (add labels and/or more elaborate caption).

Relation to Prior Work: I do not find any issues with respect to scholarship. The contribution is clearly stated in section 2.2, and a relation to prior work is clearly presented.

Reproducibility: Yes

Additional Feedback: I found this paper quite strong and very enjoyable to read. The reason for my score is primarily due to the lack of experiments showcasing the generative power of the method (see my first point under weaknesses). However, if the authors are able to provide additional results that speak towards that, I would be willing to increase my score. - For instance one could report a Monte Carlo estimate of the expected (log-)likelihood: int p(X_validation|Z) p(Z) dz for the different models. One could use the same samples Z for each model and add error bars. Below I have a few (minor) questions and suggestions. # General questions 1) I assume there is no closed form solution for figure 2? - should probably be stated, since it appears a bit misleading in its current form 2) I assume Eq. 22 is used as an objective, if so what is the value of epsilon and lambda? If the same approach is used as in the beta-VAE paper where the epsilon can be dropped, this should be mentioned and the value of lambda reported. 3) Throughout the experiment section (and figures) the results refer to the reconstruction error and divergence error. However, these terms are not actually defined anywhere. What are they? (I assume they refer to the first and second term in Eq (22), but it should be mentioned) 4) What is N_alpha? (not explicitly defined, I assume it is N_alpha=G_alpha(N_1,N_2)) 5) Eq. (22) Should the constraint not be "for all x\inX"? 6) Eq. 21 (and 22): I think in the referred work (Higgins, et al, 2017), the log is explicitly added. If the objective being optimized in this paper also includes the log, why is that not written here? 7) For training the VAE (section 3.2) what is the prior p(z)? # General suggestions 1) Somewhere on line 102-104: For completeness it would be nice to mention that p and q must have overlapping support for p^alpha * q^(1-alpha) to be normalizable for alpha in ]0, 1[. Otherwise if their supports are disjoint then for all values of x we have p(x)^alpha * q(x)^(1-alpha) = 0 as either p(x) or q(x) would equal 0. 2) Figure 3c) and 3d) are never discussed or mentioned. Since these two plots do not really support any claims made (nor contradict them) consider removing them. 3) Line 122: I do not think you should remove the prime. Either rename the original JSG or the aJSG (including the dual forms). I found it somewhat confusing when I had to go back and forth to compare things that on one page it was vanilla JSG but then aJSG on another. i) Also as suggested above I think the results should be compared against vanilla JSG as well, for which dropping the prime would not be possible. # Typos - line 207: missing word(s) around "which introduce" =========================================================================== Post Rebuttal: Thank you for providing the additional experiment (figure 1 in the response). It clarified my main concern regarding this work. My questions regarding Eqs. (20)-(21) remains unanswered however, and they should addresses in a revised version of this paper.

[Author Response · NeurIPS 2020]

We thank the reviewers for their thoughtful feedback. We are encouraged to see that all reviewers recognise our three contributions and their relation to the NeurIPS audience: introduction of our $JS^{G_\alpha}$ variant, justification of interpolation between forward and reverse KL, and application to VAE regularisation with improved empirical results. We are pleased to see that all reviewers found our paper well-written (**R3** has several detailed/useful improvements), well-related to prior work, and easily reproducible. Reviewer consensus also strongly supports the correctness of our paper, with the minor concerns addressed below.

**Generative experiments. R4**, in Fig. 1 we include experiments conducted following the review, estimating model evidence (ME) in comparison to forward KL, reverse KL and MMD (JS being intractable). ME estimates are generated by Monte Carlo estimation of the marginal distribution $p_\theta(X)$ with mean and 95% confidence intervals bootstrapped from 1,000 resamples of estimated batch evidence across 100 *test* set batches. We again emphasise here that we are not looking for SoTA, but *relative* improvement which isolates the impact of the proposed regularisation and extends our analysis of $JS^{G_{\alpha'}}$. Hopefully, these steps also address **R2**'s comment on further experimentation.

(a) MNIST      (b) FashionMNIST

Figure 1: Log model evidence compared across $\alpha$ values. The increased reconstructive power of $JS^{G_\alpha}_*$ does come at a cost to generative performance, however this trend is not consistent in the noisier (b) FashionMNIST dataset. Nevertheless, note that the reconstruction error of $JS^{G_\alpha}_*$ for (a) $\alpha > 0.8$ and (b) $\alpha > 0.6$ is still lower than the benchmarks. We also find $0.15 < \alpha < 0.4$ for $JS^{G_\alpha}$ is competitive with or better than all alternatives on both datasets.

**Correctness. R3**, L134 should indeed be $\alpha = 0$ for $JS^{G_\alpha}$ and $\alpha = 1$ for $JS^{G_\alpha}_*$ (see typo mentioned below).

**Extension of Nielsen 2019 and related work.** W.r.t. **R3**'s question about the Nielsen paper, we reverse the intermediate distribution parameterisation allowing a principled interpolation of forward and reverse KL, we simplify the subsequent closed-form loss to that needed for VAEs, and we demonstrate improved empirical performance against several baselines (application, rather than the theory of Nielsen's papers). We will amend the text to make this clearer. In this regard, we will also extend the related-work discussion, to further highlight the contributions of this work.

**Disentanglement** Although **R2** and **R4** note the question of disentanglement, we deliberately chose to leave this question for future/separate work in order to keep the message simple—interpolation between forward/reverse KL and the consequences of their associated zero-forcing/avoiding properties on VAE regularisation.

**General comments. R1**, w.r.t. improving reconstruction quality (well-known to be an issue with VAEs) we could see a future line of work using $JS^{G_\alpha}$ in nested VAEs (such as Dai and Wipf's *Diagnosing and Enhancing VAE Models*) and recent larger architectures. **R1**, yes we do have a rather important typo in Equation 19 that will be fixed. **R2**, using the proposed regularisation has no impact on complexity compared to other VAEs. Equations 23 and 24 can be vectorised and FLOPs are $\mathcal{O}(n)$ where $n$ is the size of the latent space ($\leq 32$ for our experiments). **R2**, we will make it clearer that hyperparameters for $\beta$-VAE and MMD were selected based on the most successful models in the respective papers. W.r.t **R2**'s question regarding Wasserstein Autoencoders, whilst we will consider performing detailed analysis, we expect their performance (at least WAE-MMD) to be comparable to the VAEs regularised with MMD. **R2**, we will add the limiting reconstructions of $\alpha$ from Fig. 4 to the supplementary material as this is one of the avenues we experimented with ahead of submission. **R3**, regarding better explanation of Fig. 1 (in the manuscript) being objectively in favour of $JS^{G_\alpha}$, the aim of this figure was to demonstrate that the geometric mixture of Gaussians leads to a larger integral (and therefore regularisation) contribution in areas of low probability for either distribution. The weighting term in forward and reverse KL (and therefore numerically integrated JS) will allow for very little contribution in these regions. **R4**, we will consider moving Figures 3c and 3d to the Supplementary, as space will be used elsewhere. **R3**, **R4**, thank you for the detailed minor comments/suggestions sections, they will be taken into consideration. **R1** and **R4** had suggestions for Fig. 4 which we believe to be valid. Our approach will be to reduce the number of grid entries, therefore expanding subfigure size, and add labelling/caption content to minimise ambiguity.

**Broader impact. R2**, **R3**, we accept that the broader impact statement can be expanded upon and will address this in the redraft. We will emphasise the advantages and disadvantages of using generative models, as well as synthetic (generated) data usage in various data-scarce scenarios and as a mechanism to forge real data.

[Meta-Review · NeurIPS 2020]

The paper considers the use of skewed geometric-Jensen-Shannon divergences as a replacement for the standard rate term (i.e. the divergence KL(q(z|x) || p(z)) in variational autoencoders. This paper makes 3 contributions. First, it explores the properties of the skew geometric-Jensen-Shannon divergence (JS^G_\alpha) with respect to classic Kullback-Leibler (KL) and Jensen-Shannon (JS) divergence. Second, the authors modify the JS^G_\alpha and reverse the skew parameter in the geometric distribution in order to obtain a divergence with more intuitive properties in limit cases. Finally, the authors use the JS^G_\alpha in the Variational Autoencoders (VAE) setting in replacement of the Kullback-Leibler (KL) divergence and perform experiments on several benchmark datasets to show the relevance of their method. This submission was on balance positively received by the reviewers. Reviewers found the paper clearly written and appreciate the detail with which the properties of various divergences are discussed. One point of criticism raised by reviewers is that the paper is primarily an application of divergences that were proposed in a paper by Nielsen (2019), although the authors do propose extensions to this divergence. There was some disagreement among reviewers about the strength of the experimental evaluations. One reviewer found the evaluation strong, whereas another noted that experiments only demonstrate improvements to baselines with particular choices of hyperparameters, which makes it difficult to assess the robustness of the reported results. The AC noted a couple of points where discussion of the experiments was not clear (and reviewers were not able to clarify these points). In particular: 1. A β-VAE objective with β=4 will typically have poor performance in terms of reconstruction loss, which can be improved by reducing β when this is the metric for performance. Why did the authors consider a value β<1? 2. Why do the authors consider KL(p(z) || q(z|x)) as a baseline? One would expect this to lead to a large covariance in the encoder, which once again would lead to a poor reconstruction loss. 3. Most importantly, the propose loss in equation (22) contains a hyperparameter α and a hyperparameter λ, which is analogous to the constant β in the β-VAE objective. While the authors report results for different α values, it was unclear to the AC (and after discussion also to the reviewers) how the parameter λ is chosen. Do the authors simply use λ=1 in all experiments? Based on the above points, the AC would say that it would be appropriate to do additional experiments that report results for a number of values β and λ values. These experiments are not all that computationally expensive, so this should easily be doable by camera ready. The AC is inclined to say that the contribution of this paper does not hinge on the outcome of these additional experiments. There is value in evaluating the use of these divergences in VAE objectives, regardless of their exact performance relative to other objectives. Based on this, the AC is inclined to say that this paper is just about above the bar for acceptance, provided the authors address issues with clarity and experimentation discussed in the reviews and above.